# Antibody-Drug Conjugates in Urothelial Cancer: From Scientific Rationale to Clinical Development

**DOI:** 10.3390/cancers16132420

**Published:** 2024-06-30

**Authors:** Whi-An Kwon, Seo-Yeon Lee, Tae Yoong Jeong, Hyeon Hoe Kim, Min-Kyung Lee

**Affiliations:** 1Department of Urology, Hanyang University College of Medicine, Myongji Hospital, Goyang 10475, Gyeonggi-do, Republic of Korea; 2Department of Urology, Myongji Hospital, Goyang 10475, Gyeonggi-do, Republic of Korea; photomol@hanmail.net (S.-Y.L.); urojeong@mjh.or.kr (T.Y.J.); hhkim@snu.ac.kr (H.H.K.); 3Department of Internal Medicine, Hanyang University College of Medicine, Myongji Hospital, Goyang 10475, Gyeonggi-do, Republic of Korea

**Keywords:** urothelial carcinoma, antibody-drug conjugates, ADC, enfortumab vedotin, ADC resistance mechanism

## Abstract

**Simple Summary:**

For metastatic urothelial cancer (UC), platinum-based chemotherapy and immunotherapy are used, with newer treatments like monoclonal antibodies (e.g., pembrolizumab) showing varied success. Traditional treatments often fail to provide long-term responses. Advances in molecular understanding of UC have led to targeted therapies, identifying six UC subclasses influencing treatment responses. Promising drugs include the fibroblast growth factor receptor inhibitor erdafitinib and antibody-drug conjugates (ADCs). ADCs represent a significant advancement in UC treatment, using monoclonal antibodies linked to cytotoxic agents to target cancer cells. UC is suitable for ADC therapy due to high antigen expression, enhancing efficacy while reducing systemic toxicity. Despite immune checkpoint inhibitors, advanced UC progresses rapidly with poor survival rates. Notable ADCs include enfortumab vedotin, effective alone and with pembrolizumab, and sacituzumab govitecan, showing effectiveness in studies. This review covers ADC mechanisms, mono- and combination therapies, resistance, and future perspectives, highlighting ADCs’ vital role in treating UC.

**Abstract:**

Antibody-drug conjugates (ADCs) have been a significant advancement in cancer therapy, particularly for urothelial cancer (UC). These innovative treatments, originally developed for hematological malignancies, use target-specific monoclonal antibodies linked to potent cytotoxic agents. This rational drug design efficiently delivers cancer cell-killing agents to cells expressing specific surface proteins, which are abundant in UC owing to their high antigen expression. UC is an ideal candidate for ADC therapy, as it enhances on-target efficacy while mitigating systemic toxicity. In recent years, considerable progress has been made in understanding the biology and mechanisms of tumor progression in UC. However, despite the introduction of immune checkpoint inhibitors, advanced UC is characterized by rapid progression and poor survival rates. Targeted therapies that have been developed include the anti-nectin 4 ADC enfortumab vedotin and the fibroblast growth factor receptor inhibitor erdafitinib. Enfortumab vedotin has shown efficacy in prospective studies in patients with advanced UC, alone and in combination with pembrolizumab. The anti-Trop-2 ADC sacituzumab govitecan has also demonstrated effectiveness in single-armed studies. This review highlights the mechanism of action of ADCs, their application in mono- and combination therapies, primary mechanisms of resistance, and future perspectives for their clinical use in UC treatment. ADCs have proven to be an increasingly vital component of the therapeutic landscape for urothelial carcinoma, filling a gap in the treatment of this progressive disease.

## 1. Introduction

Urothelial cancer (UC), which is predominantly found in the bladder, is the tenth most prevalent carcinoma worldwide [1]. Cystectomy is performed to treat muscle-invasive bladder cancer (MIBC), yet the recurrence rate remains high. Patients with pT3N0 tumors have a five-year survival rate between 45% and 55%, which drops to 35% to 45% for those with pT4aN0 tumors. The five-year survival rate for individuals with metastatic UC (mUC) is notably low, at only 7.7% [2]. Current treatment protocols for MIBC include the administration of cisplatin-based neoadjuvant chemotherapy followed by radical cystectomy and pelvic lymph node dissection [3]. Neoadjuvant chemotherapy can lead to a pathological complete response in 22% to 40% of cases [4,5,6]. However, the overall survival (OS) benefit from this treatment is limited to only 5% and the treatment has significant toxic side effects [5]. Treatment options have evolved for metastatic cases, including platinum-based chemotherapy and, for cisplatin-ineligible patients, a combination of chemotherapy and immunotherapy. Newer treatments involving monoclonal antibodies that include pembrolizumab, atezolizumab, and avelumab have been approved for specific patient groups with varying degrees of success. However, traditional chemotherapy and immune checkpoint inhibitors (ICIs) often fail to induce long-term responses in metastatic cases [7]. Recent developments in understanding the molecular pathways of UC have led to the development of targeted therapies. A consensus committee has identified six UC subclasses, each with unique molecular characteristics that influence treatment responses [8]. Targetable mutations identified by analysis of The Cancer Genome Atlas data have led to the development of drugs, including the fibroblast growth factor receptor inhibitor erdafitinib, which has shown promising results in advanced cases [9]. Additionally, antibody-drug conjugates (ADCs) are being explored for their ability to deliver chemotherapeutic drugs directly to cancer cells, offering a more effective and less toxic treatment option (Table 1) [10]. This review presents the latest evidence on the use of ADCs in the treatment of UC and comprehensively summarizes their mechanisms of action, clinical progress, resistance, and future perspectives.

## 2. Antibody-Drug Conjugates

In the early 20th century, Paul Ehrlich introduced the “magic bullets” concept, hypothesizing that certain compounds target specific cells to treat diseases, effectively killing cancer cells while sparing normal cells [11]. This hypothesis spurred research that has led to the identification of overexpressed antigens like human epidermal growth factor receptor 2 (HER2) in breast cancer and cluster of differentiation 20 (CD20) in B-cell lymphoma, enabling targeted cancer therapy using monoclonal antibodies (mAbs) [12]. The development of hybridoma technology in 1975 significantly advanced this field, leading to the approval of mAbs, such as avastin and trastuzumab, for various cancers [13].

The development of mAbs has revolutionized cancer treatment by accurately targeting antigens on tumor surfaces. However, the use of mAbs often falls short concerning effectiveness, especially when compared to the more lethal impact of chemotherapy on cancer cells [14]. To address this issue, a new approach, ADCs, was developed. ADCs combine the specificity of mAbs with the potency of cytotoxic drugs, thereby enhancing treatment effectiveness. ADCs are composed of mAbs directed at tumors that are linked to a potent cytotoxic drug via a carefully engineered chemical linker. This design permits the precise targeting of cancer cells and enhances the overall potency of the treatment. Additionally, the linkage of a toxic agent to a sizeable hydrophilic antibody in ADCs restricts the unintended absorption of toxic drugs by cells that do not express the target antigen, thereby broadening their therapeutic index [15].

In 2000, the United States Food and Drug Administration (FDA) approved the first ADC drug, Mylotarg^®^ (gemtuzumab ozogamicin), for treating adults with acute myeloid leukemia (AML) [15]. This approval heralded the ADC era in targeted cancer therapy. Currently, 13 ADCs have been approved globally for the treatment of both hematological malignancies and solid tumors. More than 100 ADC candidates are currently in various phases of clinical trials [16]. ADCs, with their growing range of targets and applications, are at the forefront of a new wave of targeted cancer therapies and are anticipated to eventually replace traditional chemotherapy.

### 2.1. Structures of ADC

ADCs are complex structures comprising an antibody, cytotoxic payload, and chemical linker (Figure 1). They are designed to be stable in the bloodstream, accurately target specific antigens on tumor cells, and release their cytotoxic payload to these cells. The efficacy and safety of ADCs depend on the careful selection of each component: the target antigen, antibody, cytotoxic agent, linker, and conjugation method. ADCs use mAbs to deliver potent anticancer agents directly to tumor cells, enhancing treatment effectiveness and minimizing toxicity [15]. Earlier versions of ADCs used murine antibodies. Their success was limited because of their pronounced immunogenicity and poor selectivity [17]. Technological advances have led to the development of humanized mAbs, resulting in ADCs that are more specific, effective, and less toxic. The composition of ADCs varies and is influenced by the properties of the antibody, linker, payload, and their interactions [18].

#### 2.1.1. Antibody

The development and optimization of ADCs rely heavily on the intricate selection and engineering of their components to precisely target cancer cells while minimizing their impact on normal cells. The choice of target antigens that are predominantly expressed on tumor cells rather than on healthy tissues is essential, with surface or extracellular positioning for accessibility and elimination of secretion to prevent non-targeted binding. Successful ADCs, such as adotrastuzumab emtansine and disitamab vedotin, exploit antigens like HER2, which are overexpressed in certain cancers at significantly higher levels than those in normal cells [19].

Current ADCs focus on targeting proteins that are more predominant in cancer cells across solid tumors and hematological malignancies, expanding into the tumor microenvironment (TME) to enhance efficacy and reduce drug resistance. The antibody component is crucial for targeting. This component is optimized for high affinity, efficient internalization, low immunogenicity, and prolonged plasma half-life. Humanized IgG1 antibodies are often used, reflecting their stability, immune cell engagement, and reduced immunogenicity. Despite these advancements, ADCs face challenges. One challenge is the “binding site barrier”, where excessive antigen affinity hampers tumor penetration, necessitating a balance in affinity, antibody size, and drug-to-antibody ratio for optimal therapeutic efficacy [20]. Moreover, while ideal ADCs exclusively target antigens that are absent in non-malignant tissues, practical targets, such as HER2 and TROP2, are also present in healthy tissues, leading to potential toxicities [21]. Innovations include the exploration of antibodies that recognize tumor-specific antigen variants and enhance tumor specificity. Furthermore, the internalization and turnover rates of ADCs are crucial for their efficacy, highlighting the need for careful selection and optimization of antibody backbones and binding affinities to overcome challenges that include limited tissue penetration and achieve optimal performance [22].

#### 2.1.2. Linker

The linker is essential in ADCs for connecting the antibody to the cytotoxic payload. The linker also is critical for the stability of the ADC and the controlled release of the payload, which significantly affects the therapeutic index [23]. The primary function of a linker is to keep the cytotoxic payload securely attached to the antibody during circulation in the plasma, ensuring that drugs that cannot be systemically administered are delivered effectively. If the linker is unstable, it may cause premature release of the payload into the plasma, leading to systemic toxicity and diminished therapeutic efficacy [15].

Linkers in ADCs are divided into two main subclasses: cleavable and noncleavable. Cleavable linkers release cytotoxic payloads in response to specific factors in the TME. They can be further divided into chemical cleavage linkers, such as hydrazone and disulfide bonds, and enzyme cleavage linkers, such as glucuronide and peptide bonds. Hydrazone linkers release payloads in the acidic environments of lysosomes and endosomes within cancer cells, but may also hydrolyze in plasma. Disulfide bond linkers, which are sensitive to reductive glutathione (GSH), are stable in the blood, but release their payloads in cancer cells with high GSH levels. Enzyme-sensitive linkers, such as peptide-based linkers, remain stable in the plasma and release drugs near tumors where lysosomal proteases are overexpressed. ADCs incorporating these linkers include brentuximab vedotin and sacituzumab govitecan [23].

Non-cleavable linkers, such as thioether or maleimidocaproyl groups, resist chemical and enzymatic environments in vivo and offer greater stability than cleavable linkers. They depend on the degradation of the entire antibody-linker complex for payload release, which is suitable for small molecules, as observed for ado-trastuzumab emtansine (T-DM1), which uses a thioether linker [24].

Linker selection is pivotal for ensuring payload stability and targeted release depending on the desired release mechanism and metabolic fate of the target cells [15]. Both cleavable and noncleavable linkers provide distinct advantages for cancer treatment. Non-cleavable linkers offer excellent stability in systemic circulation, requiring complete endocytosis and digestion of the antibody for payload release facilitated by cytosolic and lysosomal proteases. Cleavable linkers, which are the currently preferred in ADCs, are designed to be degraded by tumor-associated factors, enabling efficient payload release and maximizing ADC potency. However, they risk premature payload release, which leads to systemic toxicity [15].

Recent research has focused on developing more stable cleavable linkers, such as the Gly-Gly-Phe-Gly (GGFG) tetrapeptide GGFG tetrapeptide, cathepsin-responsive tripeptide linkers, and linkers cleaved by β-glucuronidase, sulfatase, phosphatase, and legumain, aiming to balance stability and efficacy for improved ADC performance [25].

#### 2.1.3. Payloads

In the design of ADCs, the cytotoxic payload, often termed the “warhead”, is a crucial component that is cytotoxic after being internalized into cancer cells. Given that only approximately 2% of an administered ADC reaches the targeted tumor sites, these payloads must be highly potent, with IC50 values in the nanomolar and picomolar ranges to ensure efficacy. The payloads must remain stable under physiological conditions and possess functional groups for antibody conjugation. Current cytotoxic payloads for ADCs include potent tubulin inhibitors, DNA-damaging agents, and immunomodulators, each selected based on the mechanism of action, stability, and potency to maximize therapeutic efficacy while minimizing systemic toxicity [15].

Tubulin inhibitors disrupt cell division by targeting microtubules, with the auristatin derivatives monomethyl auristatin E (MMAE) and MMAF promoting tubulin polymerization, and maytansinoid derivatives mertansene (DM1) and ravtansine (DM4) and tubulysins inhibiting it. For instance, adotrastuzumab emtansine employs DM1, which is the first FDA-approved ADC using maytansinoid derivatives [26].

DNA-damaging agents that are independent of the cell cycle include calicheamicins, duocarmycins, topoisomerase I inhibitors (SN-38 and DXd), and pyrrolobenzodiazepines (PBDs) [21]. These agents, such as calicheamicin in gemtuzumab ozogamicin and inotuzumab ozogamicin, and SN-38 in sacituzumab govitecan (SG), can achieve picomolar-level IC50 values, indicating high efficacy [15].

Newer payloads, such as small-molecule immunomodulators, are being explored as immune-stimulating antibody conjugates, which combine antibody targeting and immune system modulation. Examples include BDC-1001, which targets HER2 with a Toll-like receptor 7/8 agonist, and stimulator of interferon genes (STING)-agonist ADC programs, such as CRD5500 and XMT-2056 [27]. Earlier versions of ADCs attempted to use traditional chemotherapeutics, such as doxorubicin and methotrexate, which often require high doses and pose systemic toxicity risks. These attempts underscore the importance of selecting highly cytotoxic drugs that are effective at low concentrations to minimize off-target effects [28]. 

The payloads of FDA-approved ADCs and those in development, including antimitotics, DNA-damaging agents, and novel payloads, such as immunomodulators and protein-degrader-recruiting molecules, demonstrate the shift towards more targeted and potent therapeutic options. These payloads exhibit sub-nanomolar or picomolar levels of in vitro cytotoxicity that far exceed the activity of common chemotherapies [29].

Payload hydrophobicity has significant roles in ADC efficacy and toxicity, with the bystander effect allowing the eradication of heterogeneous tumors by facilitating payload diffusion into adjacent cells. However, excessive hydrophobicity can lead to rapid clearance, immunogenicity, and hepatotoxicity. Strategies to mitigate these effects include fine-tuning the drug-to-antibody ratio (DAR) and incorporating hydrophilic masking groups to balance efficacy and toxicity, highlighting complex considerations in ADC design [28].

### 2.2. Mechanism of Action of ADC and Why ADCs Are Particularly Beneficial for UC

ADCs synergistically target and efficiently eradicate cancer cells. These agents act like precision-guided “biological missiles” that are capable of precisely eliminating cancer cells, thereby enhancing treatment efficacy and minimizing off-target side effects [30]. The primary mechanism of action of ADCs is illustrated in Figure 2a. When the ADC mAb binds to the specific antigen that is uniquely expressed on cancer cells, it leads to internalization of the ADC into early endosomes. This internalization progresses to late endosomes, which eventually fuse with lysosomes. In lysosomes, lethal payloads are released either chemically or enzymatically, leading to cell death by targeting the DNA or microtubules [31]. Additionally, if the released payload can cross membranes, it may trigger a bystander effect that amplifies the effectiveness of the ADC. This effect can also modify the tumor microenvironment, potentially boosting the lethality of ADCs (Figure 3) [29].

Furthermore, the anticancer actions of ADCs are supported by their effects on antibody-dependent cellular cytotoxicity (ADCC), antibody-dependent cellular phagocytosis (ADCP), and complement-dependent cytotoxicity (CDC) [32]. The Fab segment of some ADC antibodies can attach to antigens on virus infected cells or cancer cells, whereas the Fc segment connects with Fc receptors on the surface of killer cells (such as natural killer cells and macrophages), promoting direct killing effects (Figure 2b).

Additionally, the antibody part of the ADC specifically adheres to antigens on cancer cells, blocking subsequent signaling from the antigen receptor (Figure 2c). For instance, trastuzumab targets the HER2 receptor on cancer cells and prevents the formation of heterodimers with HER1, HER3, or HER4, thus disrupting signaling pathways, such as phosphoinositide 3-kinase (PI3K) or mitogen-activated protein kinase, which are crucial for cell survival and proliferation, and lead to apoptosis [33].

Several factors contribute to the particular benefit of ADCs in UC compared to cancers of other organs. One of the primary reasons ADCs are so effective in UC is the high expression of specific cell surface proteins, such as Nectin-4 and Trop-2, on UC cells. These proteins are overexpressed in UC cells but are less prevalent in normal tissues, making them ideal targets for ADC therapy. This high expression allows for efficient targeting and uptake of ADCs by cancer cells, enhancing their therapeutic efficacy while minimizing the impact on normal cells. The unique tumor microenvironment of UC also plays a crucial role in the effectiveness of ADCs [34]. Urothelial carcinomas often have a highly vascularized and leaky tumor microenvironment, which enhances the penetration and accumulation of ADCs within the tumor [35]. This characteristic improves the delivery and efficacy of ADCs, making them particularly effective in treating UC. Additionally, UC has a high mutation burden, creating neoantigens that the immune system can recognize [36]. ADCs can exploit this immunogenic environment, as their cytotoxic payloads can induce immunogenic cell death, further stimulating anti-tumor immune responses and enhancing the overall therapeutic effect [37]. Furthermore, ADCs can be effectively combined with other treatment modalities, such as ICIs (e.g., PD-1/PD-L1 inhibitors), already used in UC. This combination can potentiate the anti-tumor immune response, offering improved outcomes for patients. The synergy between ADCs and immunotherapy represents a promising avenue for future treatment strategies.in UC [38]. Conventional therapies, such as chemotherapy and immunotherapy, often have limited efficacy and significant side effects in advanced UC [39]. ADCs provide a novel mechanism of action that can overcome resistance to these therapies. By targeting cancer cells more precisely, ADCs can achieve better outcomes with fewer adverse effects. Additionally, ADCs are designed to be more tolerable and safer than traditional chemotherapy, as they deliver cytotoxic agents directly to cancer cells, sparing normal tissues [40]. This targeted approach is particularly beneficial for UC patients, who may already have compromised health due to age or comorbidities. Improved tolerability and safety profiles make ADCs a more viable option for a broader range of patients [41].

The unique expression profiles of target antigens, the favorable tumor microenvironment, combination therapy potential, biomarker availability, and the significant unmet medical needs in UC collectively contribute to the particular benefit of ADCs in treating this UC compared to others [42]. 

## 3. Clinical Development of ADCs in UC

### 3.1. Enfortumab Vedotin (EV) Targeting Nectin-4

#### 3.1.1. Monotherapy

EV is an ADC that specifically targets nectin-4, a transmembrane protein highly expressed in UC [43]. This innovative ADC consists of a fully human mAb linked to the microtubule-disrupting agent MMAE via a protease-cleavable linker. The design of EV allows it to target the extracellular domain of nectin-4, which is overexpressed in various malignancies, including bladder, breast, colon, and cervical cancers [42]. Interestingly, in bladder cancer samples, 83% to 87% of patients express nectin-4, with expression being high in 60% of these patients [44].

The development of vedotin-based ADCs dates back to the demonstration of their superior antitumor activity compared with other auristatin family molecules [45]. As a microtubule inhibitor, vedotin hinders tubulin polymerization, induces cell cycle arrest, blocks mitosis, and triggers cell death [46]. Brentuximab vedotin, the first vedotin-based ADC, received accelerated FDA approval in 2011 for treatment-refractory Hodgkin lymphoma and systemic anaplastic large-cell lymphoma [47].

The first human study of EV was EV101, a phase-1 trial involving 155 heavily pretreated patients with locally advanced (la) UC or mUC. This study established a recommended phase 2 dose (RP2D) of 1.25 mg/kg to achieve an objective response rate (ORR) of 43%. The trial findings led to the initiation of phase II and III trials of EV in UC (Table 2) [48].

EV-201 is a global, phase II, single-arm trial that demonstrated the efficacy and safety of EV in two distinct cohorts. Cohort 1 included 125 la/mUC patients with disease progression after platinum-based chemotherapy and immunotherapy (ICI). In this cohort, EV showed an ORR of 44% (95% CI, 35.1% to 53.2%), including a 12% complete response rate (CRR) and a median duration of response (mDOR) of 7.6 months (range, 0.95–11.30+ months). The most frequently occurring side effects of the treatment included fatigue and peripheral neuropathy, both reported by 50% of the patients, followed by alopecia (49%), rash (48%), reduced appetite (44%), and dysgeusia (40%). Additionally, no severe adverse events (grade 3 or higher) affected 10% or more of the patients (Table 2) [52].

The phase 3 EV-301 trial further established the efficacy of EV by comparing it with traditional chemotherapy (docetaxel, paclitaxel, or vinflunine) in patients with la/mUC who had previously received platinum-based chemotherapy and ICI. This trial confirmed the significant improvements in progression-free survival (PFS) in patients treated with EV. A subsequent analysis after nearly two years corroborated the OS benefit of EV (median 12.9 months versus 8.9 months) [53].

Based on these results, EV received accelerated FDA approval in 2019 and full approval in July 2021. The approval was expanded to include patients ineligible for cisplatin-containing chemotherapy who received first-line ICI monotherapy [54]. Treatment-related adverse events (TRAEs) were comparable between the EV and chemotherapy arms, with EV showing higher percentages of skin reactions, peripheral neuropathy, and hyperglycemia, although these were generally mild to moderate in severity [53].

An expanded access treatment protocol was launched in Europe, highlighting a crucial safety warning regarding severe and occasionally fatal cutaneous adverse reactions associated with EV, including Stevens–Johnson syndrome and toxic epidermal necrolysis. Despite these concerns, the efficacy of EV in treating la/mUC, particularly after standard-of-care third-line treatments, has led to its rapid incorporation into clinical practice [55].

#### 3.1.2. Combinations

EV combined with pembrolizumab (EV + P) has shown promising results as a treatment for la/mUC, particularly in patients who are ineligible for cisplatin-based chemotherapy. This combination therapy, based on encouraging data from the EV-103 trial, including cohort K, has been pivotal for establishing a new standard-of-care in this domain [56].

The EV-103 phase Ib/II multicohort study (NCT03288545) has been instrumental in evaluating the efficacy of various combinations of EV for the treatment of la/mUC. In cohort K, the EV + P combination was tested as first-line therapy in 76 platinum-ineligible patients. This cohort revealed a confirmed ORR of 64.5% (95% CI, 52.7 to 75.1), with a CRR of 10.5% [43]. The same update from this cohort reported an ORR of 73.3% (95% CI, 58.1–85.4%) with a 15.6% CRR and a disease control rate of 84.4%, with an mDOR of 22.1 months and mOS reaching 26.1 months. Notably, 53% of confirmed ORRs were observed in patients with liver metastases [49]. These results highlight the effectiveness of this therapy in difficult-to-treat populations. Adverse effects were more common in the combination arm, with approximately half of the patients experiencing any-grade skin rashes and/or peripheral neuropathy. However, long-term follow-up of the EV-103 dose escalation cohort and cohort A revealed no new safety concerns, even after nearly 4 years of follow-up (Table 2) [57].

Based on these findings, the FDA granted accelerated approval in April 2023 for EV + P as a first-line therapy for cisplatin-ineligible patients with la/mUC [58]. This decision was also influenced by the impressive results of the phase III EV-302 trial (NCT04223856), which compared first-line gemcitabine plus cisplatin or carboplatin with EV plus pembrolizumab in both cisplatin-eligible and cisplatin-ineligible patients. The aim of the EV-302/KEYNOTE-A39 study was to evaluate the effectiveness of combining EV + P with platinum-based chemotherapy in patients with la/mUC who had not received prior treatment without considering their eligibility for cisplatin or expression levels of programmed death-ligand 1 (PD-L1). The participants were allocated evenly, based on their eligibility for cisplatin and PD-L1 expression, and whether they had liver metastases, to either the EV + P group, in which treatment continued until disease progression as determined by blinded independent central review, clinical worsening, intolerable side effects, or after completing the maximum number of cycles [35 for pembrolizumab], or to a regimen of gemcitabine plus cisplatin or carboplatin for up to six cycles. Demographic and baseline disease characteristics were comparable between the two groups. The median age of the participants was 69 years, with 97% showing an Eastern Cooperative Oncology Group (ECOG) performance status between 0 and 1. Approximately 75% of the patients had a primary tumor in their lower urinary tract, 54% were considered eligible for cisplatin treatment, 22% showed liver metastases, and 58% had high PD-L1 expression levels. The EV + P group showed a significant improvement in PFS from a median of 6.3 months to 12.5 months (HR: 0.45, 95% CI: 0.38–0.54, *p* < 0.0001), with benefits observed across all subgroups analyzed. OS nearly doubled in the EV + P group compared to the chemotherapy group, with median survival of 31.5 months versus 16.1 months, respectively (HR: 0.47, 95% CI: 0.38–0.58, *p* < 0.00001). This OS advantage was consistent regardless of cisplatin eligibility and was observed in both the high and low PD-L1 expression subgroups. Further subgroup analyses by age, sex, ECOG performance status, site of primary disease, and liver metastasis status favored the EV + P group. The ORR was also higher in the EV + P group (68% vs. 44%), with a notably higher CRR (29.1% vs. 12.5%, respectively). A greater percentage of patients in the chemotherapy group received further systemic treatment (66.2% vs. 28.9%), predominantly PD-1/L1 inhibitors (58.6%), and platinum-based treatments (24.9%) in the EV + P group. The frequency and nature of TRAEs were consistent with the safety profile of each drug. Serious TRAEs were reported in 27.7% of the patients in the EV + P group compared with 19.6% in the chemotherapy group. Deaths due to TRAEs occurred in 0.9% of participants in both groups. High-grade adverse events (grade 3 or above) were reported in 56% of the patients in the EV + P group and 70% of the patients in the chemotherapy group, with the most common TRAEs in the EV group being skin reactions, peripheral neuropathy, eye disorders, and high blood sugar levels (Table 2) [50]. Based on the outstanding results of this historic trial, the National Comprehensive Cancer Network recommends the EV + P combination as the primary systemic treatment for patients with la/mUC, irrespective of their eligibility for cisplatin [59].

In addition to EV-103 and EV-302, the MORPHEUS trial (NCT03869190) is another significant research effort exploring the efficacy of atezolizumab in combination with various agents, including EV. This trial comprises two stages, with the primary endpoint being ORR and a focus on safety in terms of potential overlapping toxicities [60]. A separate phase 1 trial (NCT05014139) investigated the use of intravesical infusion of EV in patients with high-risk non-muscle invasive bladder cancer who were unresponsive to Bacillus Calmette-Guérin therapy. These collective efforts underscore the ongoing exploration and potential of EV-based therapies in diverse clinical settings and stages of bladder cancer [61].

### 3.2. SG Targeting Trophoblast Cell-Surface Antigen 2 (TROP-2)

SG is an innovative ADC targeting TROP-2, a transmembrane glycoprotein overexpressed in various epithelial cancers, including UC [62]. SG consists of an anti-TROP2 antibody linked to the topoisomerase I inhibitor SN-38 via a hydrolyzable linker [63]. SG received accelerated FDA approval in April 2021 to treat patients with la/mUC who were refractory to both platinum-based and anti-PD-1/PD-L1 therapies, following encouraging results in clinical trials [64].

The TROPHY-U-01 phase II trial (NCT03547973) enrolled 113 patients with la/mUC who had previously received platinum-containing chemotherapy and ICI. The trial demonstrated an initial ORR of 27%, with CRR in 5.4% of patients, and an mDOR of 7.2 months. With longer follow-up, the ORR remained high at 28%, including in patients with heavily pretreated la/mUC, those who had previously received EV, and those with disease progression after previous platinum-based therapy (Table 2) [51].

Additional cohorts in the TROPHY-U-01 trial further explored the efficacy of SG. Cohort 3 included 41 platinum-refractory la/mUC patients who had not received an ICI and were treated with SG plus pembrolizumab. These patients had an ORR of 41% (95% CI, 26.3 to 57.9; 20% CRR), median DOR was 11.1 months (95% CI, 4.8 to not estimable [NE]), and median PFS was 5.3 months (95% CI, 3.4 to 10.2). The median OS was 12.7 months (range, 10.7-NE). The trial also examined SG in combination with various other therapies, such as cisplatin, avelumab, and the anti-PD-1 antibody zimberelimab, revealing the potential of SG using diverse therapeutic combinations [65].

The mechanism of action of SG involves a humanized mAb against TROP-2 conjugated with SN-38, the active metabolite of irinotecan, which inhibits topoisomerase I, which is essential for DNA integrity [66]. TROP-2’s overexpression in UC and its minimal presence in normal tissues makes it an ideal target for SG [67].

Regulatory approval of SG for la/mUC is contingent on the verification of its clinical benefits in the ongoing phase III TROPiCS-04 (NCT04527991) trial. In this trial, patients with la/mUC following platinum-based chemotherapy and ICI are receiving SG or single-agent chemotherapy with the primary endpoint of OS [65].

### 3.3. Other ADCs

Over the past two decades, significant advancements have been made in the treatment of UC, particularly with the development of ADCs targeting HER2 [68]. HER2, a member of the epithelial growth factor receptor family, is overexpressed in a notable percentage of UC cases, particularly cisplatin-resistant tumors [69]. Despite initially inconsistent results with anti-HER2 agents, recent developments in ADCs have shown promising outcomes [70].

The incidence of genomic alterations in the erbB-2 gene encoding HER2 was 6% to 17% in MIBC or la/mUC [71]. Traditional strategies targeting HER2, such as trastuzumab and tyrosine kinase inhibitors (afatinib, neratinib, and lapatinib), have not significantly improved the outcomes in patients with la/mUC. However, newer ADCs, such as disitamab vedotin and trastuzumab deruxtecan (T-DXd) are promising. These ADCs, which consist of a humanized anti-HER2 mAb linked to cytotoxic agents, can effectively target tumor cells with low levels of HER2 expression and exert cytotoxic effects on neighboring tumor cells via a bystander effect [72].

Clinical trials have demonstrated the efficacy of these ADCs in treating HER2-positive la/mUC. The phase II RC48-C011 trial of disitamab vedotin reported an ORR of 26% in patients with HER2-negative or low la/mUC, with 68% achieving stable disease. In heavily pretreated patients with HER2-positive la/mUC, the ORR was 50.5%. Combining disitamab vedotin with the anti-PD-L1 antibody toripalimab resulted in an ORR of 75%, which increased to 100% in patients with HER2 IHC scores of 2+/3+ and PD-L1-positive disease. However, TRAEs are common and include increased serum alanine aminotransaminase and aspartate aminotransferase levels, peripheral sensory neuropathy, and hypertriglyceridemia [73].

Similarly, a phase Ib/II study of T-DXd combined with nivolumab in HER2-positive la/mUC patients who had previously received platinum-based therapy showed an ORR of 37% and a mDOR of 13 months. The most common TRAEs are nausea, fatigue, and vomiting, with a significant incidence in drug-related interstitial lung disease/pneumonitis [72].

Beyond ADCs, novel therapeutic classes such as Bicycle Toxin Conjugates (BTCs) are emerging. BT8009, a nectin-4-targeting BTC linked to MMAE, showed promising results in a phase I/II trial, with an objective response in 50% of patients with la/mUC and a clinical benefit of over 16 weeks in 75% of patients [74].

The integration of ADCs and BTCs into systemic therapies for la/mUC represents a significant shift in the treatment landscape. The combination of ADCs with anti-PD-1 or anti-PD-L1 antibodies is particularly promising and has led to the design of several ongoing phase III trials. These developments underscore the emergence of HER2 as an important therapeutic target for UC, and offer new hope for patients with this challenging disease.

## 4. Resistance to ADCs

Because the structures of ADCs are complex and their mechanism of action consists of multiple steps, resistance can occur at any level, beginning with antigen expression and recognition through internalization and degradation for cytotoxic drug release and apoptotic regulation [40]. Early evidence suggests that resistance to ADCs can occur via several mechanisms, including prevention of antibody binding, alteration of ADC processing and internalization, and loss of payload efficacy [42]. A more accurate understanding of these mechanisms may provide additional insights into the intrinsic mechanisms of action of the drug and accelerate the development of predictive biomarkers of efficacy. Figure 4 illustrates the mechanisms of resistance and strategies for overcoming resistance to ADC.

### 4.1. Antigen-Related Resistance

ADCs target specific antigens. However, resistance can develop due to changes in the recognition of these antigens by mAbs. Loss or downregulation of target antigen expression, gene mutations that make the antigen less recognizable, or selection of non-antigen-expressing cells within a heterogeneous cell population are key resistance factors [40].

This reduction in antigen expression affects the efficacy of ADCs, leading to diminished antibody binding and payload release. Clinical trials, such as the EMILIA and ASCENT trials, showed that patients with higher levels of the target antigens (HER-2 mRNA in EMILIA, Trop-2 in ASCENT) had better responses to ADC treatments [21,75]. Studies by Loganzo et al. and Coates et al. indicated that long-term exposure to ADCs can lead to drug resistance, as evidenced by reduced antigen protein levels or loss of antigen expression [76]. Tumor heterogeneity in antigen expression significantly affects ADC efficacy, as demonstrated in the KRISTINE and ZEPHIR trials, where patients with higher heterogeneity showed less response or faster treatment failure [77,78]. Resistance mechanisms include truncated forms of the antigen ectodomain as observed with trastuzumab [79]. The masking and isolation of antigens by extracellular matrices or the presence of ligands, such as heregulin, can further modulate ADC sensitivity [80]. 

Concerning nectin-4 in UC, Klümper et al. showed that the downregulation of target antigens during disease progression can be associated with resistance to therapies like EV [81]. Monitoring antigen expression could be crucial for predicting responses to treatment and guiding therapy choices. Future research needs to explore whether multimodal approaches, such as bispecific antibodies targeting multiple antigens, can overcome these resistance mechanisms [82].

### 4.2. Payload-Related Resistance

Resistance to chemotherapy, including treatments involving ADCs, often arises through various cellular mechanisms. One prominent cause is the elimination of cytotoxic agents used in ADCs by ATP-binding cassette transporters [83]. These transporters, particularly multidrug resistance-1 (MDR-1), remove ADC payloads, such as auristatin analogs and maytansinoids, from cells, leading to drug resistance. Extended exposure to these drugs can result in the selection of cell clones that overexpress MDR-1, further contributing to resistance [42].

Another significant resistance mechanism involves mutations in the molecular targets of ADC payloads. For example, mutations in topoisomerase-1 (TOP1), which is targeted by the SN-38 payload in certain ADCs, can reduce the susceptibility of tumors to treatments. Additionally, tumor cells may develop resistance to the payload independent of antigen resistance [84].

This type of resistance was initially observed in non-Hodgkin lymphoma tumors, where switching from an auristatin-based to anthracycline-based payload improved the ADC response [85]. In support, Takegawa et al. described that cells resistant to ADC T-DM1 retained normal HER2 expression but showed increased expression of the ABC transporters ABCC2 and ABCG2 [86]. Importantly, blocking these transporters reversed the resistance, highlighting enhanced efflux as a key mechanism of resistance. These insights into resistance mechanisms are crucial for developing strategies to enhance the efficacy of ADC therapies by overcoming drug resistance pathways [87].

Payload diversification has emerged as a strategy for enhancing the therapeutic effects of ADCs. For example, SKB-264, a TROP-2-targeted ADC that uses a belotecan derivative instead of the traditional topoisomerase-1 inhibitor payload has show promise in ongoing clinical trials [88].

Conjugation chemistry and DAR are crucial for ADC design. ADCs can be linked through cysteine or lysine residues, with an optimal DAR range reported to be 2–4 for cysteine-linked and 3–4 for lysine-linked ADCs [89]. The effectiveness of conjugation methods varies; some studies found lysine-conjugated ADCs to be more effective, whereas others showed better results with cysteine-based site-specific conjugation [90].

Although the DAR is a key factor in ADC efficacy, there is no universally optimal DAR. Studies indicate that ADCs with very high DARs (approximately 9–10) may have faster clearance and reduced efficacy than those with lower DARs [91]. However, other aspects, such as the conjugation site and drug loading, are also influential.

### 4.3. Internalization and Trafficking Pathways-Related Resistance

The efficacy of ADCs is optimized through the internalization of the antibody into cancer cells via endocytosis. ADCs commonly use routes like clathrin- or caveolae-mediated endocytosis, and clathrin-caveolin-independent endocytosis [40]. Different endocytosis routes are regulated by specific proteins, such as adaptor protein, dynamin, epsin, and phosphatidylinositol bisphosphate, each playing a unique role in cellular processes [92]. However, these endocytosis mechanisms limit payload efficacy. For instance, internalization of TDM-1 into caveolin-1 coated vesicles has been linked to decreased sensitivity and increased insensitivity to treatment [42].

In vitro models of T-DM1 resistance have varied; two models showed resistance through reduced HER2 expression, while N87-TM cells retained normal HER2 levels and internalized trastuzumab-ADCs into caveolin-1 coated vesicles. The findings indicate the significance of the microenvironment and enzymes in antibody catabolism and delivery, with a preference for cleavable linkers in TDM [31,93]. In HER2-positive breast cancer, endophilin A2 enhances HER2 internalization and sensitivity to trastuzumab-based therapy. Knockdown of SH3GL1, which encodes endophilin A2 in tumor cells, reduced internalization and suppressed T-DM1-mediated cytotoxicity [79].

### 4.4. Other Types of ADC Resistance

ATP-binding cassette transporters, especially MDR1, are crucial for chemotherapy resistance, expelling drugs from cells, and leading to multidrug-resistant tumors. ADC payloads, often substrates for these transporters, face resistance due to this [94]. Kovtun et al. suggested that hydrophilic linkers in ADCs, such as PEG4Mal, improve the efficacy against MDR1-expressing tumors [95]. Cell cycle dynamics indicate that actively cycling leukemic cells are more susceptible to chemotherapy. T-DM1 resistance is linked to cyclin B, which is involved in cell cycle transition [40]. Sabbaghi et al. found that manipulating cyclin B1 levels could sensitize T-DM1 resistant cells [96]. Activation of signaling pathways, including PI3K/AKT/mammalian target of rapamycin, affects ADC sensitivity, and trastuzumab resistance is associated with phosphatidylinositol-4,5-bisphosphate 3-kinase mutations or phosphatase and tensin homolog deletions [79]. However, T-DM1 remained effective in such cases, as observed in the EMILIA trial [97]. Wnt/β-catenin pathway activation, particularly via Wnt3 overexpression, is linked to trastuzumab resistance [98]. Finally, apoptotic dysregulation affects ADC sensitivity. Overexpression of the apoptosis-related proteins B-cell lymphoma 2 and BCL-XL correlates with resistance to ADCs, such as gemtuzumab, ozogamicin, or brentuximab vedotin, in hematological cancers [40].

## 5. Strategies to Overcome ADC Resistance

Despite the significant progress in ADC technology, overcoming resistance to cancer treatment remains a formidable challenge. This difficulty stems from various complex mechanisms, some of which are not fully understood. A prevalent form of drug resistance is enhanced expulsion of drugs attributed to the overexpression of drug efflux pumps [85].

Takegawa et al. illustrated how T-DXd successfully countered T-DM1 resistance in HER2-positive gastric cancer cells [86]. This breakthrough was further validated by the DESTINY-Breast02 phase III trial, which compared T-DXd with chemotherapy in patients with HER2-positive metastatic breast cancer resistant to T-DM1 [99]. In parallel, research has shown that altering the linker in ADCs to increase hydrophilicity, as demonstrated by Sulfo-SPDB-DM4 and PEG4Mal, offers another effective strategy against resistance, particularly in tumors positive for MDR1 [40].

The complexity of tumors, especially those with cells expressing low antigen levels, poses significant challenges. Investigations such as the study by Golfier et al. on anetumab ravtansine have highlighted the strong antitumor activity of ADCs, even against variably expressing tumors [100]. Similarly, Li et al. revealed that increasing the rate of payload release in ADCs can lead to heightened potency and a more pronounced bystander effect, offering another avenue for enhancing the effectiveness of ADCs [101].

Innovative antibody designs have promise in combating resistance. Bispecific or biparatropic mAbs, such as the novel biparatropic ADC targeting two distinct HER2 epitopes, have shown efficacy in treating T-DM1-resistant and low-HER2-expressing tumors [40]. Andreev et al. explored bispecific ADCs connecting HER2 and the prolactin receptor to enhance the binding and internalization of these ADCs, leading to improved anticancer results [102]. This is further supported by early results from a phase I trial of zanidatamab zovodotin (ZW49), which indicated a tolerable safety profile and promising response rate [103]. SHR-A1811, a next-generation ADC, exhibited remarkable outcomes in a phase I trial, particularly in patients with HER2-positive breast cancer previously treated with T-DM1 who were resistant to other HER2-targeted ADCs [104].

In addition to these innovative designs, advances in conjugation chemistry have enabled the creation of dual-drug ADCs that deliver two distinct cytotoxic agents, offering another strategy to overcome resistance [105]. Dual-drug ADCs utilize branched chemical linkers with orthogonally masked cysteine residues to homogeneously conjugate two drugs, such as MMAE and MMAF, to an antibody [106]. MMAE effectively penetrates cell membranes and causes bystander effects, while MMAF is potent against MDR+ cells without bystander effects [107]. The combination of these drugs has demonstrated enhanced and synergistic antitumor activity in mouse models, overcoming resistance where single-drug ADCs failed [106]. Further advancements include enzymatic conjugation methods that produce highly homogeneous dual-drug ADCs with defined DARs, showing exceptional antitumor activity in treatment-resistant cancers. These dual-drug ADCs, including those combining different payload classes like anti-microtubule agents and Toll-like receptor agonists, have shown promising synergistic effects and immunological memory. However, not all dual-drug ADCs achieve meaningful synergy, highlighting the importance of selecting complementary mechanisms of action and optimizing DAR. Ongoing research aims to fully harness the potential of dual-drug ADCs to overcome resistance and improve outcomes in heterogeneous and refractory tumors, further enhancing the landscape of ADC-based therapies [33,108].

Besides individual drug enhancements, combination therapies may enhance treatment efficacy. The KATE2 study explored the combination of the atezolizumab ICI with T-DM1, with potential benefits in a PDL1-positive subgroup [109]. This approach is not limited to immunotherapy; ongoing evaluations of tyrosine kinase inhibitor combined with T-DM1 or T-DXd are underway to determine their potential synergistic effects [110]. Additionally, targeting heregulin or neuregulin 1 (NRG1), which activates the HER3 receptor, has been effective in combination with trastuzumab emtansine (T-DM1) and pertuzumab, inhibiting the NRG1/HER3 signaling pathway, leading to increased tumor suppression and extended tumor regression [111]. These advancements reflect the dynamic and intricate nature of research in combating ADC resistance and offer hope for more effective cancer treatments.

## 6. Future Perspectives and Conclusions

Advancement of ADCs in cancer therapy, particularly for UC, has led to significant progression in oncological treatments. ADCs deliver cytotoxic agents directly to tumor cells, offering a targeted approach with potent antitumor activity. This mechanism affects individual cells and modulates the TME, enabling systemic immune response modulation. ADCs combined with ICIs have shown synergistic effects, enhancing the immune response against cancer cells by promoting cell death, releasing tumor antigens, and alleviating immunosuppression in the TME.

Clinical trials such as EV-103 and EV-302 are exploring the efficacy and safety of ADC and ICI combinations in la/mUC, reflecting the evolving treatment landscape. Emerging studies have highlighted the ability of ADCs to stimulate antitumor immunity through mechanisms like immune cell death, suggesting potent antitumor responses when combined with ICIs. There are many ongoing trials of ADCs alone or in combination with established therapies for UC (Table 3).

The development of novel ADCs aims to overcome resistance mechanisms, reduce toxicity, and improve targeting capabilities. Innovations, such as bispecific antibodies, dual-payload ADCs, and strategies for enhancing tumor penetration, promise to address drug resistance and improve treatment efficacy. The therapeutic potential of ADCs is vast. However, challenges, including drug resistance, tumor heterogeneity, and the risks of TRAEs, must be addressed. 

Site-specific antibody modification for ADCs is a technology designed to enhance the efficacy and safety of these conjugates by attaching drugs to specific sites on antibodies [112]. This technique targets particular amino acid residues or precise locations on the antibody to ensure that the drug is delivered more accurately to target cells, such as cancer cells [37]. Key methods for site-specific antibody modification in ADCs include engineered cysteine conjugation, native amino acid modification, site-specific enzymatic conjugation, and click chemistry [113]. These techniques offer several advantages, including high selectivity, as the drug can precisely reach target cells and minimize side effects, increased stability by conjugating at specific sites to enhance the stability of the ADC, and enhanced efficacy by increasing the effective concentration of the drug to maximize therapeutic effects. This technology holds significant promise for applications in cancer therapy and the treatment of various diseases, with extensive research and development ongoing to further refine and expand its use [31,114]. Site-specific antibody modification methods, particularly enzyme-mediated site-specific modifications, are a current trend in the field, with an increase in preclinical ADCs employing these techniques [115]. While industrialization has posed challenges, especially with the introduction of enzyme tag residues, recent reports have successfully overcome these issues [116,117]. This progress is crucial for refining ADC technology, ensuring more effective and safer therapeutic options in the biopharmaceutical landscape.

The future of ADCs involves combining this platform with other intervention strategies, such as immunomodulation and targeting of undruggable proteins, to implement multimodal cancer treatments. Unlocking the full potential of ADCs will require comprehensive patient stratification and biomarker identification to enable personalized medicine, particularly given the heterogeneous nature of many solid tumors.

Over the past few decades, academia–industry collaborations have led to the development of various ADCs, revolutionizing cancer treatments for many patients. Fourteen ADC have been successfully launched, and others show promising clinical performance. Future advancements hinge on identifying new antigens and antibodies, developing optimal payloads, and designing effective linkers that are essential for next-generation ADCs. Further breakthroughs in targeted therapy are expected.

ADCs offer a novel approach for administering conventional cytotoxic therapies, reducing toxicity, and enhancing therapeutic efficacy, especially in diseases with specific targetable antigens. ADCs are a viable option for treating la/mUC, with drugs like EV approved and others under investigation. Future research should explore their use in early disease stages and in combination with other innovative agents to maximize efficacy and overcome resistance mechanisms.

ADCs have become a recognized part of the UC anticancer arsenal. Their complexity poses developmental challenges. However, the number of approved ADCs is expected to increase, addressing the unmet medical needs for both common and rare diseases. New ADCs have shown promising efficacy and acceptable toxicity profiles in patients with UC. The optimal duration of therapy, treatment beyond complete response, retreatment efficacy, and the real synergy between ADCs and ICIs need to be clarified. The use of ADCs in UC treatment realizes a long-held dream of preferentially targeting cancer cells over healthy cells, with ongoing research optimizing their application, timing, and positioning ADCs as a cornerstone of future cancer therapy.

## Figures and Tables

**Figure 1 cancers-16-02420-f001:**
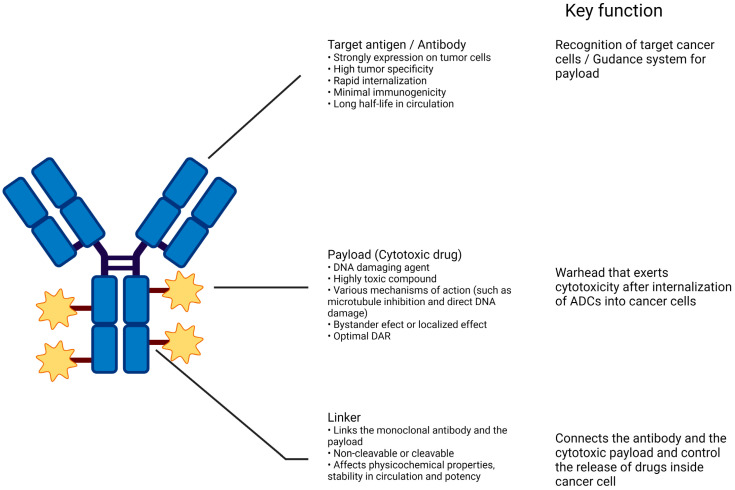
Structure of conventional ADCs. The composition of antitumor ADCs integrates three essential components: a monoclonal antibody that targets an antigen uniquely present on the surface of tumor cells, ensuring targeted delivery; a covalent linker that controls the release of the therapeutic substance within the tumor cells rather than in circulation; and a cytotoxic payload that prompts apoptosis in tumor cells by attacking critical cellular structures like DNA, microtubules, and topoisomerase 1. Created with BioRender.com.

**Figure 2 cancers-16-02420-f002:**
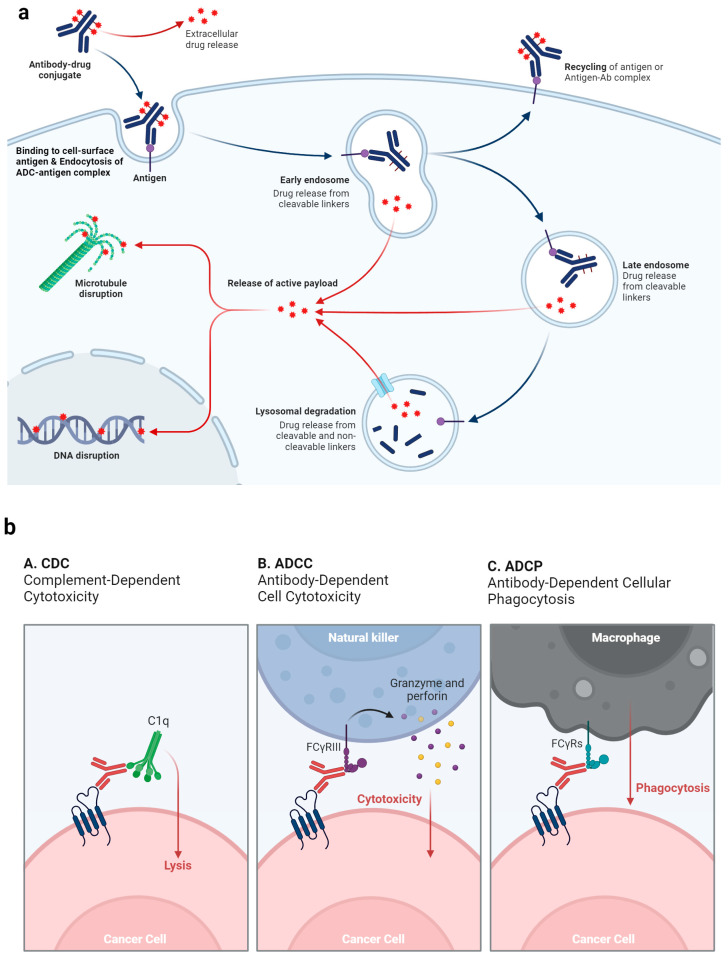
Various strategies by which ADCs target and eliminate cancer cells. (**a**) Overview of mechanism of action of ADCs. (**b**) ADCs interact with immune effector cells to provoke a response against tumors, including CDC, ADCC, and ADCP effects. (**c**) The antibody component of the ADC maintains its functional profile, allowing it to interfere with the targeted cell’s function, suppress further signaling pathways, and thereby inhibit the proliferation of the tumor. Created with BioRender.com.

**Figure 3 cancers-16-02420-f003:**
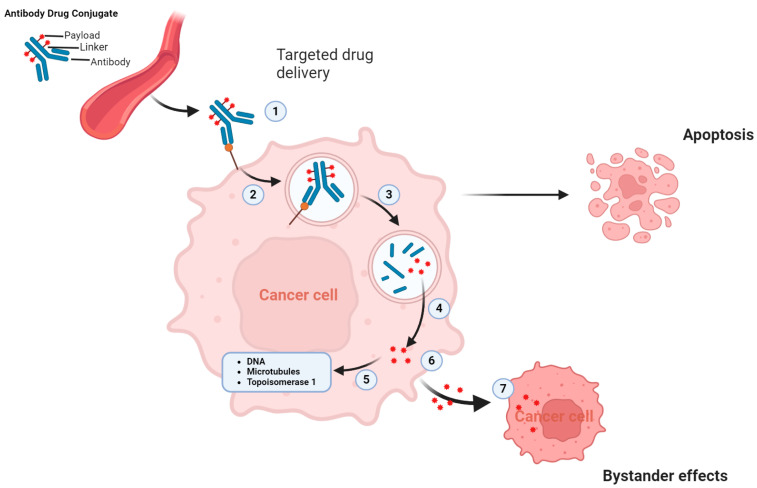
Mechanism of action of conventional ADCs for killing cancer cells via different approaches. The process of ADC-induced cytotoxicity requires sequential key steps: (1) binding to specific antigen, (2) internalization of the ADC–antigen complex, (3) lysosomal degradation of the antibody, (4) release of payload within the cytoplasm, (5) interaction of the cytotoxic drug with intracellular target. A fraction of the payload may be released in the extra-cellular environment (6) where it can be taken up by adjacent cells, (7) causing the bystander effect. Created with BioRender.com.

**Figure 4 cancers-16-02420-f004:**
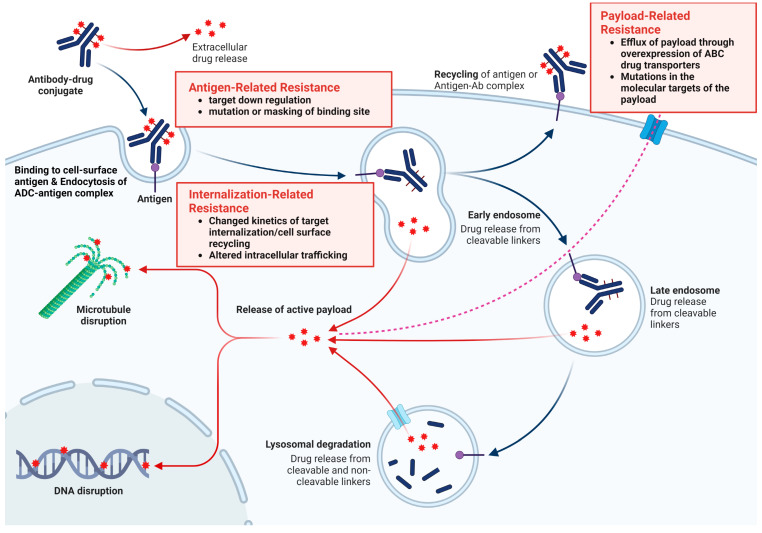
Mechanisms of resistance and strategies to overcome resistance to ADC. Created with BioRender.com.

**Table 1 cancers-16-02420-t001:** Summarizes ADCs that have been studied in UC and their mechanism of action.

ADC	Brand Name	Year Approved	Target Receptor	Linker	Payload	Payload Action	DAR
Enfortumab vedotin	Padcev	2019	Nectin-4	Enzyme-cleavable	MMAE	Microtubule inhibitor	3.8
Sacituzumab govitecan	Trodelvy	2020	Trop2	Acid-cleavable	SN-38	Topoisomerase-DNA complex Inhibitor	7.6
Disitamab vedotin	Aidexi	2021	HER2	Protease cleavable	MMAE	Microtubule inhibitor	4
Trastuzumab deruxtecan	Enhertu	2019	HER2	Enzyme-cleavable	Deruxtecan	Topoisomerase-DNA complex Inhibitor	8

Abbreviations: MMAE, monomethyl auristatin E; MMAF, monomethyl auristatin F; ADCs, Antibody–drug conjugates; DAR Drug-to-antibody ratio; HER2, Human epidermal growth factor receptor 2; MMAE, Monomethyl auristatin E; SN-38, 7-ethyl-10-hydroxycamptothecin.

**Table 2 cancers-16-02420-t002:** Trials of ADC in urothelial cancer.

Drugs	Trial	NCT Number	Study Phase	Regimen	No. of Pts.	Study Population	Primary End Points	ORR,% (95% CI)	mDOR, Months (95% CI)	mPFS, Months (95% CI)	mOS, Months (95% CI)	Ref.
Enfortumab Vedotin (EV)	EV-101	NCT02091999	I	EV	155	la/mUC	safety/tolerability and pharmacokinetics	43			12.3	[48]
	EV-201	NCT03219333	II	EV	125 (cohort 1)	la/mUC, previously treated with ICIs	ORR	44% (35.1 to 53.2)	7.6 (6.34 to NA)	5.8 (4.93 to 7.46)	12.4 (9.46 to 15.57)	
	EV-103	NCT03288545	Ib/II	EV and pembrolizumab	45 (cohort k)	la/mUC, 1L cisplatin-ineligible	safety/tolerability	73.3% (95% CI, 58.1 to 85.4%)	22.1 months (8.38, -)	12.7 months (6.11, -)	26.1 months (15.51, -)	[49]
	EV-302	NCT04223856	III	EV and pembrolizumab	442	la/mUC, 1L	PFS, OS	67.7% (95% CI, 63.1 to 72.1)	NA	12.5 months (10.4 to 16.6)	31.5 (25.4 to NE)	[50]
Sacituzumab Govitecan (SG)	TROPHY-U-01	NCT03547973	II	SG	113 (cohort 1)	la/mUC, previously treated with Plat and ICIs	ORR	27% (19.5 to 36.6)	7.2 (4.7 to 8.6 months)	5.4 months (3.5 to 7.2 months	10.9 months (9.0 to 13.8 months)	[51]

Abbreviations: CI, confidence interval; EV, enfortumab vedotin; ICI, immune checkpoint inhibitor; la/mUC: locally advanced or metastatic urothelial carcinoma; mDOR, median duration of response; mPFS, median progression-free survival; mOS, median overall survival; NCT, national clinical trial; No, number; ORR, objective response rate; Plat (platinum), cisplatin or carboplatin; Pts, patients; Ref, reference; SG, sacituzumab vedotin.

**Table 3 cancers-16-02420-t003:** Ongoing Trials of ADC in Urothelial Cancer.

**A.** Monotherapy.						
**Drugs**	**Trial**	**NCT number**	**Study Population**	**Study Phase**	**No. of Pts.**	**Primary or Co-Primary** **End Points**
EV	EV-103, cohort H	NCT03288545	Neoadjuvant in cisplatin-ineligible MIBC naive to systemic therapy	I/II ^b^	457	pCRR
EV	EV-103, cohort L	NCT03288545	Perioperative, previously untreated, cis-ineligible MIBC	I/II	50	pCRR
SG	SURE-01	NCT05226117	Neoadjuvant in cisplatin-ineligible MIBC naive to systemic therapy	II	56	pCRR
SG vs. CTX	TROPiCS-04/Immu-132–13	NCT04527991	La/mUC refractory to platinum and anti-PD-1/PD-L1 Therapies (in China)	III ^a^	696	OS
T-DXd	DestinyPanTumor02, Arm2	NCT04482309	HER2-positive solid tumor including bladder cancer	II ^b^	268	ORR
T-DXd	DestinyPanTumor01	NCT04639219	HER2-positive, unresectable or metastatic solid tumors refractory to prior therapy with limited options.	II ^b^	102	ORR
Abbreviations: AE, adverse event rate or incidence; BCG, bacillus calmette guerin; CTX, standard chemotherapy, often physician’s choice; EV, enfortumab vedotin; Gem, gemcitabine; ICI, immune checkpoint inhibitor; LA, laboratory abnormalities; la/mUC, locally advanced or metastatic urothelial cancer; MIBC, muscle-invasive bladder cancer; MTD, max tolerated dose; NCT, national clinical trial; ORR, objective response rate; OS, overall survival; P, pembrolizumab; pCRR, pathologic complete response rate; PFS, progression-free survival; Plat (platinum), cisplatin or carboplatin; SG, sacituzumab vedotin; T-DM1, trastuzumab emtansine; T-DXd, trastuzumab deruxtecan; Taxane, paclitaxel or docetaxel.—a Randomized; b Represents total aggregated in multicohort or basket trial.						
**B.** Combination Therapy.						
**Drugs**	**Trial**	**NCT number**	**Study Population**	**Study Phase**	**No. of Pts.**	**Primary or Co-Primary** **End Points**
EV + P	EV-103, Cohort A	NCT03288545	First line in platinum ineligible la/mUC refractory to prior therapies.	I/II	45	AE, ORR
EV + P	EV-103, Cohort B	NCT03288545	Second line in la/mUC refractory to platinum therapies.	I/II ^b^	457	AE, ORR
EV + Cis	EV-103, Cohort D	NCT03288545	First line in platinum-eligible la/mUC	I/II ^b^	457	AE, ORR
EV + Carbo	EV-103, Cohort E	NCT03288545	First line in cisplatin-ineligible, carboplatin-eligible la/mUC	I/II ^b^	457	AE, ORR
EV + Gem	EV-103, Cohort F	NCT03288545	First and second line in platinumineligible la/mUC refractory to prior therapies	I/II ^b^	457	AE, ORR
EV + Plt + P	EV-103, Cohort G	NCT03288545	First line in platinum-eligible la/mUC	I/II ^b^	457	AE, ORR
EV + P	EV-103, Cohort J	NCT03288545	Neoadjuvant in cisplatin-ineligible MIBC naive to systemic therapy	I/II ^b^	457	AE, ORR, pCRR
EV + P vs. Gem + Plat	EV-302	NCT04223856	First line in cisplatin-eligible la/mUC; arm A [3-week cycles of EV 1.25 mg/kg IV on days 1 and 8 (no maximum cycles) + P 200 mg IV (maximum 6 cycles)] or arm B [Gem + Cis or Carbo (maximum 6 cycles)]	III ^a^	990	PFS, OS
EV + P + Si	516–003, Cohort 9	NCT03606174	La/mUC refractory to platinum and ICI therapies	I/II ^b^	425	ORR
EV + At	MORPHEUS-UC	NCT03869190	PD1-expressing la/mUC refractory to platinum therapy	IB/II ^b^	735	ORR, pCRR
EV + P vs. P vs. Cystectomy	KEYNOTE-905/EV-303	NCT03924895	Neoadjuvant therapy or surgery alone in cisplatin-ineligible MIBC; arm A (neoadjuvant P 200 mg IV Q3W up to 3 cycles followed by RC + PLND and adjuvant P 200 mg IV Q3W up to 14 cycles), arm B (RC + PLND followed by observation), or arm C (neoadjuvant EV 1.25 mg/kg + P 200 mg IV Q3W up to 3 cycles followed by RC + PLND and adjuvant EV + P up to 6 cycles and adjuvant P 200 mg IV Q3W up to 8 cycles)	III ^a^	857	pCRR, PFS
EV + P vs. CTX	Keynote-B15/EV-304	NCT04700124	Neoadjuvant in cisplatin-eligible MIBC; arm A (4 cycles of neoadjuvant EV 1.25 mg/kg + P 200 mg IV Q3W followed by 5 cycles of adjuvant EV 1.25 mg/kg + 13 cycles of adjuvant P 200 mg IV Q3W after RC + PLND) or arm B (4 cycles of neoadjuvant Gem 1000 mg/m^2^ chemot + Cis 70 mg/m^2^ IV Q3W followed by observation after RC + PLND)	III ^a^	784	pCRR, PFS
EV + Du + Tr vs. EV + Du	VOLGA	NCT04960709	Neoadjuvant in cisplatin-ineligible systemic therapy-naive MIBC; arm A (Du 1500 mg day 1 + T 75 mg day 1 + EV 1.25 mg/kg Days 1 & 8); arm B (Du 1500 mg day 1 + EV 1.25 mg/kg Days 1 & 8; or arm C no neoadjuvant treatment (SoC)	III ^a^	830	pCRR, EFS
SG + Cis + Av	Trophy-U-01, Cohort 4	NCT03547973	Platinum naive and cisplatin-eligible la/mUC with responders receiving Avelumab maintenance	II ^b^	643	ORR, PFS
SG + Cis + Zi	Trophy-U-01, Cohort 4	NCT03547973	Platinum naive and cisplatin-eligible la/mUC with responders receiving Avelumab maintenance	II ^b^	643	ORR, PFS
SG 1 Zi or Av or Zi	Trophy-U-01, Cohort 5	NCT03547973	la/mUC maintenance therapy following Gem-Cis	II ^b^	643	ORR, PFS
SG or SG + Zi or SG + Zi + Do or GC	Trophy-U-01, Cohort 6	NCT03547973	Cisplatin-ineligible, treatment-naive la/mUC	II ^b^	643	ORR, PFS
SG + At	MORPHEUS-UC	NCT03869190	PD1-expressing la/mUC refractory to platinum therapy	IB/II ^b^	735	ORR, pCRR
DV	Keynote-D78/RC48G001, Cohort A	NCT04879329	HER2-positive, platinum-refractory la/mUC	II ^b^	270	ORR
DV	Keynote-D78/RC48G001, Cohort B	NCT04879329	HER2 low expressing, platinum refractory la/mUC	II ^b^	270	ORR
DV vs. DV + P	Keynote-D78/RC48G001, Cohort C	NCT04879329	HER2-positive, platinum eligible, treatment-naive la/mUC	II ^b^	270	ORR
DV 1 To vs. Gem + Plat	RC48-C016	NCT05302284	HER2-positive platinum-eligible treatment-naive la/mUC; arm A [2-week cycles of DV 2.0 mg/kg IV + 2-week cycles of To 3.0 mg/kg IV (until confirmed disease progression, unacceptable toxicity, or voluntary withdrawal)] or arm B [Gem + Cis or Carbo Q3W (maximum 6 weeks or until investigator assessed loss of clinical benefit, unacceptable toxicity, investigator or participant decision to withdraw from therapy, or death)]	III ^a^	452	PFS, OS
T-DM11At or Taxane	TDM4529 g	NCT00781612	Extension at close of parent study in eligible patients	II ^b^	720	AE, DLT
Abbreviations: AE, adverse event rate or incidence; At, atezolizumab; Av, avelumab; BCG, bacillus calmette guerin; Ca, cabozantinib; Carbo, carboplatin; Cis, cisplatin; CTX, standard chemotherapy, often physician’s choice; DCR, disease control rate; DLT, dose-limiting toxicity; Do, domvanalimab; Du, durvalumab; DV, disitamab vedotin; EFS, event-free survival; EpCAM, epithelial cell adhesion molecule; Er, erdafitinib; EV, enfortumab vedotin; every 3 weeks, Q3W; Gem, gemcitabine; ICI, immune checkpoint inhibitor; Ip, ipilizumab; Intravenously, IV; La/mUC, locally advanced or metastatic urothelial cancer; MIBC, muscle-invasive bladder cancer; MTD, maximum tolerated dose; Ni, nivolumab; NMIBC, non-muscle invasive bladder cancer; OM, oportuzumab monatox; ORR, objective response rate; OS, overall survival; P, pembrolizumab; pCRR, pathologic complete response rate; PFS, progression-free survival; Plat (platinum), cisplatin or carboplatin; SG, sacituzumab vedotin; Si, sitravatinib; SoC, standard of care; T, trastuzumab; T-DM1, trastuzumab emtansine; T-DXd, trastuzumab deruxtecan; Taxane, paclitaxel or docetaxel; TD, trastuzumab duocarmine; Tis, tislelizumab; To, toripalimab; Tr, tremelimumab; Tu, tucatinib; TV, tisotumab vedotin; Zi, zimberelimab.—a Randomized; b Represents total aggregated in multicohort or basket trial.						

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
