# Peer review of "Antibody-Drug Conjugates in Urothelial Cancer: From Scientific Rationale to Clinical Development"

_cancers, 2024, doi:10.3390/cancers16132420_

Round 1

Reviewer 1 Report

Comments and Suggestions for Authors

Antibody-drug conjugates (ADCs) have been a significant advancement in cancer therapy, particularly for urothelial cancer (UC). This review article well-organize and introduce the latest evidence on the use of ADCs in the treatment of UC and comprehensively summarizes their mechanisms of action, clinical progress, resistance, and future perspectives. This is well written. In order to understand the latest evidence of ADC for UC, I think that we should read this review article. I would like to send some comment on this article. On current, there are many clinical trials and the standard therapy renews year by year. For this review article, I think that now is the best time to be published.

Major issue

I think that Tables are most important for this review article. Therefore, I think that authors should check them carefully, again. I think that the following Phase III trials, EV-303, EV-304, and VOLGA are described more accurately.

EV-303: EV+P+RC vs P+RC vs RC alone

EV-304: EV+P+RC vs CTX+RC

VOLGA: EV+Du+RC vs EV+Du+RC vs RC alone

RC: radical cystectomy + pelvic lymph node dissection

EV303 trial:

Patients were randomly assigned to arm A (neoadjuvant pembro 200 mg intravenously [IV] every 3 weeks [Q3W] up to 3 cycles followed by RC + PLND and adjuvant pembro 200 mg IV Q3W up to 14 cycles), arm B (RC + PLND followed by observation), or arm C (neoadjuvant EV 1.25 mg/kg + pembro 200 mg IV Q3W up to 3 cycles followed by RC + PLND and adjuvant EV + pembro up to 6 cycles and adjuvant pembro 200 mg IV Q3W up to 8 cycles).

EV304 trial:

Approximately 784 patients will be randomly assigned 1:1 to receive either 4 cycles of neoadjuvant EV + pembrolizumab followed by 5 cycles of adjuvant EV + 13 cycles of adjuvant pembrolizumab after RC+PLND or 4 cycles of neoadjuvant cisplatin-based chemotherapy followed by observation after RC+PLND. Neoadjuvant and adjuvant pembrolizumab 200 mg + EV 1.25 mg/kg will be administered intravenously every 3 weeks (Q3W), and neoadjuvant chemotherapy will consist of gemcitabine 1000 mg/m2 + cisplatin 70 mg/m2 Q3W.

VOLGA trial:

Patients will be randomized to 3 arms to receive 3 cycles of neoadjuvant therapy Q3W as follows: (Arm 1) D (1500 mg day 1) +T (75 mg day 1) +EV (1.25 mg/kg Days 1 & 8); (Arm 2) D (1500 mg day 1) +EV (1.25 mg/kg Days 1 & 8); or (Arm 3) no neoadjuvant treatment (SoC). A safety run-in study is included. 

Minor comments

1                 In the Figure, I prefer noun to passive voice of verb.I prefer “abundant expression on tumor cells”or “strong expression on tumor cells” to “strongly expressed” in Figure 1.

2                 Please use I, II, or III instead of 1, 2, or 3 in Study phase of Table 2. Abbreviations are apparently insufficient.

3                 In Table 1, I think that payload action is not different EV from DV. Please check them. What is vc-PABC Valyl-citrullinyl-p-aminobenzyloxycarbonyl? What is CL2A A cleavable complicated PEG8- and 635 triazole-containing PABC-peptide-mc linker? Please check them in Table 1?

4                 pCR or pCRR? Please unify them. pCRR: pathological complete response rate in Table 3. What is EFS of VOLGA?

Reviewer 2 Report

Comments and Suggestions for Authors

This review covers ADC mechanisms, mono-and combination therapies, resistance, and future perspectives.

The review is well-written and clear. The figure's quality is good.

Minor:

Check typos

Introduction: line 49-53 please consider citing doi: 10.3390/jpm14020212. 

Reviewer 3 Report

Comments and Suggestions for Authors

This review summarizes the mechanisms of action, clinical development, resistance mechanisms, and future perspectives of antibody-drug conjugates (ADCs) in urothelial carcinoma (UC). This work is comprehensive and compelling, and I believe it has the potential for publication. However, I would like to suggest some revisions to enhance the manuscript's depth and originality.

  1. Discussion on the Relevance of ADCs to UC: Although this review extensively covers ADCs, the manuscript lacks a discussion from the perspective of why ADCs are particularly beneficial for UC. Many recent reviews have summarized ADCs well, but to add originality and depth to this review, it is crucial to focus on the relationship between ADCs and UC. Please include a detailed discussion on the significance of using ADCs for UC, addressing their advantages, potential challenges, and why they are particularly suited for this type of cancer.

  2. Future Perspectives - Site-Specific Antibody Modification: In the section on future perspectives, I recommend including a discussion on site-specific antibody modification methods. This is a current trend in the field, especially enzyme-mediated site-specific modifications, which have seen an increase in preclinical ADCs. While industrialization has posed challenges with the introduction of enzyme tag residues, recent reports have overcome these issues. Incorporating these methods and their advancements will provide a comprehensive view of future ADC developments. I also suggest citing recent reviews on tag-free antibody modification methods to support this discussion.
    https://doi.org/10.1002/slct.202203753

  3. Discussion on Dual ADCs: Dual ADCs are gaining attention as a means to overcome resistance to payloads. Although still in the preclinical stage, it is important to touch on Dual ADCs that incorporate payloads with different mechanisms of action. Including this discussion will highlight cutting-edge strategies and provide a forward-looking perspective on overcoming resistance in ADC therapies.

Round 2

Reviewer 1 Report

Comments and Suggestions for Authors

In order to understand the latest evidence of ADC for UC, I think that we should read this review article. Authors kindly reply my request. Thank you very much.

Reviewer 3 Report

Comments and Suggestions for Authors

Accept in present form